# Molecularly Imprinted Magnetic Nanocomposite Based on Carboxymethyl Dextrin for Removal of Ciprofloxacin Antibiotic from Contaminated Water

**DOI:** 10.3390/nano13030489

**Published:** 2023-01-25

**Authors:** Golnaz Heidari, Fereshte Hassanzadeh Afruzi, Ehsan Nazarzadeh Zare

**Affiliations:** 1School of Chemistry, Damghan University, Damghan 36716-45667, Iran; 2Catalysts and Organic Synthesis Research Laboratory, Department of Chemistry, Iran University of Science and Technology, Tehran 13114-16846, Iran

**Keywords:** carboxymethyl dextrin, magnetic molecularly imprinted polymer, adsorption, ciprofloxacin

## Abstract

Broad-spectrum antibiotics from the fluoroquinolone family have emerged as prominent water contaminants, among other pharmaceutical pollutants. In the present study, an antibacterial magnetic molecularly imprinted polymer (MMIP) composite was successfully fabricated using carboxy methyl dextrin grafted to poly(aniline-*co*-meta-phenylenediamine) in the presence of Fe_3_O_4_/CuO nanoparticles and ciprofloxacin antibiotic. The characteristics of obtained materials were investigated using FTIR, XRD, VSM, TGA, EDX, FE-SEM, zeta potential, and BETanalyses. Afterward, the MMIP’s antibacterial activity and adsorption effectiveness for removing ciprofloxacin from aqueous solutions were explored. The results of the antibacterial tests showed that MMIP had an antibacterial effect against *Escherichia coli*, a Gram-negative pathogen (16 mm), and *Staphylococcus aureus*, a Gram-positive pathogen (22 mm). Adsorption efficacy was evaluated under a variety of experimental conditions, including solution pH, adsorbent dosage, contact time, and initial concentration. The maximum adsorption capacity (Q_max_) of the MMIP for ciprofloxacin was determined to be 1111.1 mg/g using 3 mg of MMIP, with an initial concentration of 400 mg/L of ciprofloxacin at pH 7, within 15 min, and agitated at 25 °C, and the experimental adsorption results were well-described by the Freundlich isotherm model. The adsorption kinetic data were well represented by the pseudo-second-order model. Electrostatic interaction, cation exchange, π-π interactions, and hydrogen bonding were mostly able to adsorb the majority of the ciprofloxacin onto the MMIP. Adsorption–desorption experiments revealed that the MMIP could be retrieved and reused with no noticeable reduction in adsorption efficacy after three consecutive cycles.

## 1. Introduction

Pharmaceutical contamination of the aquatic environment has become a major global problem. The pharmaceutical sector, among other industries, is reported to use 22% of all industrial freshwater, creating wastewater that is either disposed of directly or is not appropriately treated before disposal [1,2]. More than 600 active pharmacological compounds have been found in water bodies across the world today, including drinking water [3,4]. Pharmaceutical active compounds can have negative impacts on humans, aquatic life, and the environment, even at a minute level. Of those, antibiotics have been classified as developing contaminants because of their constant intake and endurance in waterbodies, and because of their low biodegradability [5,6]. Additionally, antibiotic overuse generates massive amounts of antibiotic wastewater, which introduces bacterial resistance into natural environments. One significant class of widely used antibiotics with undetectable biodegradability is the quinolone antibiotic.

Ciprofloxacin has excellent antibacterial activity and is a crucial fluoroquinolone antibiotic that is frequently used to treat illnesses with large production. Through interactions with topoisomerases II and IV in bacteria, it can prevent DNA replication [7]. Ciprofloxacin is most typically seen in wastewater treatment plant effluents because it is too chemically stable to be fully degraded [8]. Bacterial resistance to fluoroquinolone antibiotics, which are normally present at low concentrations, has grown as a result of increased exposure to these drugs in recent years [9,10]. Ciprofloxacin has greater stability in soil and wastewater systems and a higher aqueous solubility under different pH levels. Several physical and chemical approaches, including liquid extraction (such as HPLC) [11,12], adsorption [4], photodegradation [13], ozonation chemical oxidation [14], and membrane techniques [15,16,17], have been developed to treat antibiotic wastewater by eliminating different organic components. Among those, adsorption as a physicochemical treatment method has received a lot of interest for removing ciprofloxacin from wastewater because it is low in cost, has no undesirable byproducts, highly efficient, and easy to use in large-scale industrial applications [18,19,20,21].

Compared to other commonly used materials such as resin, activated carbon, and biochar, the molecularly imprinted polymer-based (MIP) unconventional adsorption technique provides a viable solution for the selective removal of certain target contaminants [22,23]. In MIPs, by applying the molecular imprinting technique, precisely aligned three-dimensional cavities with the template molecule of the polymer substrates are produced [24,25,26]. A technique to create pattern-shaped holes in material matrices with predetermined selectivity and high affinity is called molecular imprinting. The notion of producing specific substrate recognition sites in a matrix by using the casting technique is one of the most naturally occurring nature-imitating methods. Structural predictability, specific target identification, and practicability are all characteristics of highly cross-linked MIPs. They have good physical stability, and could be used in severe environments, such as high pressures and temperatures, extreme pH, and organic solvent reactions [27]. Thus, chemical, mechanical, and thermal stability, ease of use and low cost in preparation, adaptability to a wide range of target molecules, and great resistance to organic solvents are some intriguing benefits of MIPs [22].

Dextrin, a biodegradable and biocompatible material, is a polysaccharide with a low molecular weight composed of d-glucose units connected by α-(1,4) or α-(1,6) glycoside linkages. It is frequently used as an adhesive in the food and textile industries and is produced by hydrolyzing starch or glycogen with heat, acids, and enzymes [28]. On the other side, inorganic nanoparticles distributed in polymers provide a unique function in improving the mechanical, thermal, and chemical characteristics and antibacterial activity of polymer matrices significantly. Therefore, the development of MIPs employing nanoparticles, such as copper oxide and iron oxide, can improve the abovementioned benefits. As a result, the polymer’s surface area will rise, resulting in increased exposure of the MIP cavities and a reduction in the resistance to mass transfer. Copper oxide (CuO) nanoparticles have special qualities and have been utilized to disinfect substances, liquids, and human tissue [29,30,31,32]. Additionally, the magnetic characteristic of iron oxide nanoparticles will make it easier to remove the MIP from the sample once the analyte has been isolated, eliminating the requirement for the centrifugation stage.

Moreover, the size and geometry of the holes created by structural monomers or crosslinkers, as well as the chemical interactions between the template and the functional monomer, determine the specificity of the MIPs. Because of this, selecting the functional monomer that will be utilized for MIP synthesis is essential for achieving successful outcomes. Non-covalent interactions between the functional monomer and template, such as hydrophobic or ionic contact, dipole–dipole and π–π interactions, and hydrogen bonding, are commonly used to generate MIPs. As a result, it is chosen based on the functional groups present in the chemical structure of the molecule utilized as a template. Chemical groups in Ciprofloxacin, such as secondary and tertiary amines, and a carboxylic acid, can interact chemically with MIP cavities. Consequently, herein, MMIP was developed in three steps: self-assembly, copolymerization, and template removal. Afterward, MMIP can interact with ciprofloxacin and remove it from water through electrostatic interaction, cation exchange, π–π interactions, and hydrogen bonding.

## 2. Experimental

### 2.1. Materials and Instruments

Aniline (Merck) was purified by double distillation before being used. All chemicals were purchased from Merck Company (Germany) and used as received. Fourier transform infrared spectroscopy (FTIR) spectroscopy was performed using a Bruker Tensor 27 spectrometer (Bruker, Karlsruhe, Germany). A vibrating sample magnetometer (Daghigh Kavir, Kashan, Iran) was used to conduct the vibrating sample magnetometer (VSM) measurement. The LENSES STAPT-1000 calorimeter (Linseis, STA PT1000, Selb, Germany) was used to conduct the thermogravimetric analysis (TGA) in a nitrogen environment at a heating rate of 10 °C/min in the range of 0–800 °C. The ultraviolet-visible (UV-vis) spectra (Cecil 5000 UV-vis, UK) spectrophotometer were used to measure ciprofloxacin concentration in the solutions. X-ray diffraction (XRD) patterns were recorded on a Riga kuD/Max-2550 powder diffractometer with a scanning rate of 5°/min. A field emission scanning electron microscope coupled with an energy dispersive X-ray spectrometer (FESEM/EDX) (MIRA 3-XMU, Tescan, Kohoutovice, Czech Republic) was used to analyze the surface morphology of the materials (Hitachi S4160, Tokyo, Japan).

### 2.2. Preparation of Carboxymethyl Dextrin (CMD)

In a one-necked flask, 0.5 g dextrin was dissolved in 15 mL isopropanol and magnetically stirred for 15 min to achieve a homogenous solution. 0.6 g of NaOH dissolved in 2 mL of isopropanol was added to the solution and stirred for one hour at 30 °C. Following that, 0.68 g of monochloroacetic acid diluted in 2 mL of isopropanol was added to the solution and stirred for one hour at 40 °C. The reaction mixture was then doubled in volume with cold methanol, and the resultant yellow precipitate was separated using filter paper. Finally, the product was rinsed with ethanol and dried in a vacuum oven (Figure 1A).

### 2.3. Preparation of Iron Oxide Nanoparticles

The iron oxide nanoparticles were prepared to utilize the in situ co-precipitation approach described in the literature [33]. 5.84 g of FeCl_3_·6H_2_O and 2.17 g of FeCl_2_.6H_2_O were added into deionized water (100 mL) and stirred for 2 h under a nitrogen atmosphere at 80 °C. Then, 10 mL of NH_4_OH solution (25%) was added to the mixture, resulting in a black suspension. The sediment was finally separated by a magnet and washed multiple times with water, and dried at 70 °C for 24 h. 

### 2.4. Preparation of Copper Oxide Nanoparticles

The copper oxide nanoparticles were prepared via the solvothermal method based on the reported procedure [32]. In a round-bottom flask, 0.36 g of copper acetate and 100 mL of deionized water were added, followed by the addition of 2 mL acetic acid and stirring for 1 h at 100 °C. After that, 0.8 g of NaOH was added to the mixture to adjust the pH level to 6–7, resulting in a black precipitate. It was then centrifuged, rinsed three times with water, and dried for 24 h at 50 °C.

### 2.5. Preparation of Fe_3_O_4_/CuO Hybrid Nanoparticles

In a round-bottom flask, 1 g of iron oxide and 0.5 g of copper oxide were mixed with 50 mL of water. The mixture was subjected to ultrasonic waves for 30 min. The pH of the mixture was then adjusted to 8 using NaOH solution and stirred for 30 min at 80 °C. Eventually, the product was separated using a magnet, washed with water, and dried at 80 °C. 

### 2.6. Fabrication of MMIP

In a three-necked flask, 1.87 g carboxyl methyl dextrin was dissolved in 90 mL distilled water and magnetically stirred for 15 min to make a homogenous solution. The solution was then purged with N_2_ gas for 30 min and then aniline (1 g in 10 mL of HCl 1M) and meta-phenylenediamine (mPDA) (1 g) were added to the solution. Fe_3_O_4_/CuO hybrid nanoparticles (10 wt.% to dextrin and monomers) were dispersed by an ultrasonic bath for 30 min before being added to the previous reaction solution. Then, to start the graft copolymerization of aniline and mPDA, an aqueous solution of ammonium persulfate (2.5 g in 15 mL water) and ciprofloxacin (0.1 g in water or methanol) was introduced. Afterward, to remove ciprofloxacin, the reaction contents were washed with a mixture of methanol/acetic acid (8:2) for 16 h and distilled water (30 mL) three times. Lastly, the product was washed with ethanol to remove water and methanol traces (Figure 1B).

### 2.7. Antibacterial Study

To investigate the antagonistic behavior of the produced materials, the agar well diffusion test was used as the principal test. Two bacterium broth media were inoculated into tubes containing tryptic medium or Mueller Hinton broth, and the tubes were warmed at 37 °C for 4 h. At the end of the incubation time, the optical density of each solution was measured at 625 nm. With a final cell density of around 1.5108 CFU/mL, the absorbance of the injected media was adjusted to 0.08–0.1. These microbial solutions were added to the medium with a sterile cotton swab. A sterilized glass tube was then used to transfer the broth medium into the wells. The wells were filled with samples at a concentration of 600 mg/mL. The plates were maintained at −4 °C temperature for 4 h before adding the samples to disperse the loaded samples and inhibit the growth of growing bacteria. After 24 h of heating at 37 °C, the diameter of the inhibitory zone (mm) was measured.

### 2.8. Adsorption Experiments

The ability of MMIP to remove ciprofloxacin from aqueous solutions was tested in several ways. The effect of critical parameters on adsorption capacity was examined, including solution pH, adsorbent amount, contact time, and initial ciprofloxacin concentration in an aqueous solution. The pH was adjusted from 4 to 10 using HCl (0.1 N) and NaOH (0.1 N). After that, the optimal adsorption conditions were determined by testing different amounts of MMIP (3–20 mg), contact periods (5–30 min), and initial ciprofloxacin concentrations (100–400 ppm). The isotherms were studied by varying the concentration of the adsorbing systems (keeping other parameters at optimized levels) in different concentrations. For kinetics studies, pseudo-first-order and pseudo-second-order models were selected. For studying kinetics, adsorption systems were kept at different contact times. The experimental tests were carried out three times, and an average of the results was provided. A UV-visible spectrometer was used to measure the ciprofloxacin concentration. Ciprofloxacin’s adsorption capacity and efficiency onto MMIP was determined using Equations (1) and (2), respectively.
R% = ((C_i_ − C_e_)/C_i_) × 100(1)
Q_e_ = ((C_i_ − C_e_)/m) × V(2)
where C_i_ and C_e_ are the initial and the equilibrium concentration of ciprofloxacin in the aqueous solutions (mg/L), respectively. m is the weight of MMIP adsorbent (g) and V is the solution volume (L).

### 2.9. Desorption and Reusability

Reusability experiments of MMIP for ciprofloxacin antibiotic adsorption were conducted for three adsorption–desorption cycles. For this, ciprofloxacin had been adsorbed onto MMIP, then the ciprofloxacin that had been adsorbed onto MMIP was immersed in ethanol and stirred at room temperature for four hours to investigate its desorption and recoverability. The MMIP was then separated using a magnet. Subsequently, using a UV-visible spectrophotometer, the amount of released ciprofloxacin in the elution medium was determined. The desorption percentage was calculated using the following equation.
%D = A/B × 100(3)
where A is the mg of the ciprofloxacin desorbed to the elution medium and B is the mg of the ciprofloxacin adsorbed on the MMIP. 

## 3. Results and Discussion

### 3.1. Characterization

**FTIR**: The functional groups of the prepared materials were evaluated using FTIR spectroscopy. Figure 2A displays the FTIR spectra of dextrin, CMD, Fe_3_O_4_/CuO, and MMIP. The C-H and O-H stretching vibrations are represented by the absorption bands at 2924 cm^−^^1^ and 3432 cm^−^^1^ in the FTIR spectrum of dextrin, respectively. The stretching vibrations of the C-O-C (glycosidic bridge) and C-O-H bonds are also attributed to the absorption bands in the range of 1050–1060 cm^−^^1^ and 1000 cm^−^^1^, respectively. In comparison to dextrin, CMD shows a characteristic band at 1654 cm^−^^1^ that is related to the carbonyl groups in the carboxylate. Two characteristic bands at 590 cm^−^^1^ (the stretching vibrations of the Cu-O bond) and 670 cm^−^^1^ (the stretching vibrations of surface Fe-O-Fe) are observed in the FTIR spectra of Fe_3_O_4_/CuO nanoparticles. The stretching vibrations of the C-H, hydroxyl group, and C=O carboxylate group are discernible in the FTIR spectrum of MMIP at 607 cm^−^^1^, 2927 cm^−^^1^, and 3399 cm^−^^1^, respectively. In addition, carbonyl groups in CMD, quinoid, and benzoid rings in poly(aniline-co-meta phenylene diamine) are responsible for bands at 1651 cm^−^^1^, 1503 cm^−^^1^, and 1470 cm^−^^1^. Furthermore, the absorption band in the region of 3410 cm^−^^1^ is related to the stretching vibrations of -N-H and -NH_2_, and –OH. The successful synthesis of MMIP was confirmed by the presence of distinctive bands associated with Dex, CMD, and Fe_3_O_4_/CuO with minute shifts in the FTIR spectrum.

**XRD**: XRD analysis was used to investigate the crystal structure of Dex, CMD, Fe_3_O_4_/CuO, and MMIP (Figure 2B). The XRD pattern of Dex showed a semicrystalline nature with distinct peaks corresponding to the dextrin in the range of 2θ = 15–25°. In diffractograms of CMD, the decrease in crystallinity is caused by the replacement effect of COO^−^ groups. This is because a higher level of replacement causes a significant loss of crystallinity [34]. The XRD pattern of Fe_3_O_4_/CuO nanoparticles revealed a crystalline nature with several distinct peaks at 2θ = 31°, 36°, and 40°. More amorphous areas can be seen in the MMIP XRD pattern, which suggests that the presence of Fe_3_O_4_/CuO nanoparticles had no impact on the level of crystallinity. Therefore, it could be concluded that the incorporation of Fe_3_O_4_/CuO throughout the polymeric network was successful. 

**EDX**: The chemical composition of the CMD, Fe_3_O_4_/CuO, and MMIP were determined using EDX spectroscopy (Figure 3A). The presence of Fe, Cu, and O elements in the EDX spectrum of Fe_3_O_4_/CuO nanoparticles and O and C elements in the EDX spectrum of CMD confirmed their chemical composition. The presence of the N peak in the MMIP’s EDX spectrum, along with C, O, Fe, and Cu, verified the presence of the N-contained copolymer and the successful fabrication of the MMIP. 

**FESEM**: The morphology of samples was investigated using field emission scanning electron microscopy (FESEM). The FESEM micrographs of CMD, Fe_3_O_4_/CuO, and MMIP are shown in Figure 3B. CMD FESEM micrograph revealed that carboxymethylation triggered granular disintegration in the Dex [35]. In the FESEM image of the Fe_3_O_4_/CuO sample, a semi-spherical structure with a large concentration of nanoparticles can be noticed. Polymer morphology is widely recognized to be influenced by a variety of parameters such as acidity, monomer-to-initiator ratio, synthesis method, temperature, and others [36]. As a result, every parameter change might result in a change in the structural morphology of the polymer. And here, the MMIP image shows that it is mostly made up of unstructured geometry and has a tendency to agglomerate. The aggregated phenomenon might be caused by the nanoparticles’ high surface energy. Comparing the surface morphology of the MMIP to that of the CMD reveals notable differences, which appear to be caused by the incorporation of Fe_3_O_4_/CuO nanoparticles throughout the CMD matrix and also can be ascribed to graft copolymerization of aniline and meta-phenylenediamine and its matrix.

**VSM**: The magnetic characteristics of Fe_3_O_4_/CuO nanoparticles and MMIP were investigated employing a vibrating sample magnetometer (VSM) (Figure 4A). The Fe_3_O_4_/CuO nanoparticles’ saturation magnetization value was determined to be at 57.19 emu g^−^^1^, whereas the MMIP was observed at 16.49 emu g^−^^1^. The inclusion of Fe_3_O_4_/CuO nanoparticles between MMIP polymer chains could be responsible for its magnetic capability. The decrease in the Ms value may also be related to the MMIP’s lower magnetic component content. Nevertheless, since copolymers may lessen the agglomeration of magnetic nanoparticles, it can be inferred that the nonmagnetic copolymer is present in the nanocomposite matrix. Furthermore, the S-shaped (M-H) magnetization curve of Fe_3_O_4_/CuO nanoparticles and MMIP polymer without hysteresis loop specifies (i.e., zero values of remanence (Mr) and coercivity (Hc)) approves their superparamagnetic feature. 

**TGA**: The thermogravimetric analysis (TGA) of CMD, Dex, Fe_3_O_4_/CuO, and MMIP was carried out in an inert atmosphere at temperatures ranging from 50 to 800 °C (Figure 4B). Dex and CMD thermograms revealed three and four weight reduction stages, respectively. The initial weight losses in Dex and CMD were roughly 5% and 7%, respectively, due to water evaporation. Dehydration and complete breakdown of the Dex and CMD are related to the second (67%) and third (23%) loss phases in DEX, respectively, and the second (34%), third (15%), and fourth (22%) weight loss steps in CMD, respectively. Moreover, when the temperature rose, the Dex had the quickest rate of decomposition. This made it clear why the CMD’s thermal stability outperformed that of the Dex [37]. The elimination of moisture from the sample results in a slight weight loss of around 3% in the TG thermogram of Fe_3_O_4_/CuO. The residual weight of the nanocomposite at 800 °C was around 48 wt%, which was larger than that observed for the CMD (22%) and Dex (5%). Therefore, the addition of Fe_3_O_4_/CuO nanoparticles can enhance the MMIP’s thermal stability by serving as a thermal barrier to preserve the copolymer from heat degradation. 

**BET**: Brunauer–Emmett–Teller (BET) analysis was employed to study the textural properties, porous structure, and specific surface area of the Fe_3_O_4_/CuO and MMIP. The specific surface area of the Fe_3_O_4_/CuO and MMIP was measured at about 17 m^2^/g and 46 m^2^/g, respectively. N_2_ adsorption–desorption isotherms for both samples are shown in Figure 5, with a type-IV profile and a hysteresis loop between adsorption and desorption, indicating their mesopores structure. Therefore, the high surface area and porous structure of the fabricated adsorbent, MMIP, facilitates the diffusion and adsorption of antibiotic molecules. 

**Zeta potential:** Zeta potential is a physical property that is exhibited by any particle in suspension, macromolecule, or material surface. To determine the surface charge of MMIP, the zeta potential technique was conducted. The measured zeta potential of MMIP at pH 6, 7, and 8 can be observed in Table 1. The zeta potential of MMIP in the pH range from 6–8 displaying an increase in negative potential as conditions become more basic, which may result from the deprotonation of its functional groups (hydroxyl and carboxyl groups).

**Antibacterial study**: One of the key ideas for the survival of people, animals, and plants is the control of the growth of microorganisms in the environment. Antibiotic-resistant bacteria are quickly turning into a difficult medical problem in modern society. Even the most modern medications cannot effectively treat bacterial infections that are resistant to two or more types of antibiotics [38]. In the post-antibiotic time, it is vitally necessary to discover new and alternative methods to fight multi-drug-resistant bacterial diseases. Therefore, the preparation of antimicrobial biomaterials, which seems to be a potential strategy for the decontamination of bacteria-contaminated water is of considerable interest. Additionally, various efforts have been made to create polymeric materials with possible antimicrobial activity. In this respect, the antibacterial activity of CMD, Fe_3_O_4_/CuO, and MMIP was assessed against *Escherichia coli* (*E. coli* is a Gram-negative bacteria) and *Staphylococcus aureus* (*S. aureus* is a Gram-positive bacteria). The results of the inhibition of bacteria were correlated to the commonly used antibiotics gentamicin and chloramphenicol (Table 2 and Figure 6). While the Fe_3_O_4_/CuO nanoparticles showed strong antibacterial action against both *E. coli* and *S. aureus* bacteria, CMD did not exhibit inhibition against them. The formation of radical oxygen species by the released Fe and Cu ions, which damages the bacterial cell membranes, is thought to be the cause of the Fe_3_O_4_/CuO nanoparticles’ antibacterial properties [39]. The observed synergic effect in the antibacterial activity of MMIP could be because of the presence of Fe_3_O_4_/CuO and poly(phenylenediamine) and aniline in the polymer structure [40,41,42]. 

### 3.2. Optimization of the Effective Parameters on the Ciprofloxacin Adsorption 

Ciprofloxacin adsorption onto MMIP depends on several parameters, including pH, adsorbent dosage, contact duration, and initial concentration. These factors can significantly increase the effectiveness of antibiotic adsorption. Thus, they have been studied carefully.

#### 3.2.1. Solution pH

pH is regarded as one of the most important external factors in the adsorption process because variations in pH can influence not only the surface charge of the adsorbent but also the existing speciation of adsorbates in solution. In order to determine the impact of solution pH on the efficiency of ciprofloxacin’s adsorption onto MMIP, the pH of the aqueous solutions was changed from 4 to 11. The results are displayed in Figure 7A. Based on reported information in the literature, ciprofloxacin shows two pKa values of 5.90 and 8.89 corresponding to the amine and carboxylic acid groups [7]. Ciprofloxacin has a cationic form because of the protonation of the amine group below pH 5.9; it shows zwitterionic properties between pH 5.9 and 8.9 and also has an anionic form when the pH is greater than 8 due to the deprotonation of carboxylic acid. Considering that the adsorbent exhibits a negative surface charge anywhere in the pH range of 1 to 5.59 of the solution, electrostatic interactions and cation exchange (with the protonated amine groups of ciprofloxacin and MMIP) often become active. It could be inferred from the evaluation that π–π interactions, hydrogen bonds, and van der Waals forces, particularly in the neutral state, were the major influences on how ciprofloxacin bound to the surface adsorption sites of the substance. According to the pH graph, the adsorption capacity rises as the pH rises to 7. The maximum adsorption capacity, 342.28 mg/g, was obtained at pH 7. However, since ciprofloxacin was in anionic form after pH 8.9, it was repulsed by the negative surface of MMIP. In this case, a decrease in adsorption capacity was observed after this pH value (pH = 9–11).

#### 3.2.2. Adsorbent Dosage

To determine the relationship between the quantity of the MMIP adsorbent and its capacity for ciprofloxacin adsorption, adsorption experiments were conducted in the presence of various concentrations of the MMIP at the ideal pH. According to Figure 7B, 3 mg of MMIP was adequate for the adsorption of ciprofloxacin in an aqueous solution. The adsorption capacity decreased when the dose was increased from 3 mg to 20 mg, and thereafter there was minimal change. Therefore, 3 mg of MMIP was employed for additional research. The equilibrium between the adsorbent surface and the ciprofloxacin ions accounts for this. The high adsorption capacity at a lower adsorbent dosage is related to the high quantity of antibiotic that is accessible for adsorption [18]. This could be ascribed to the rise in active sites that are accessible, allowing for maximal ciprofloxacin absorption. 

#### 3.2.3. Contact Time

The impact of contact time, a key parameter, can be used to guarantee and identify the maximum adsorption efficiency. To verify adsorption effectiveness, several contact times from 5 to 30 min were explored. The adsorption efficiency increases from 5 to 15 min, starts to decline from 15 to 20 min, and then shows a small, inconsequential increment, as seen in Figure 7C. The reason for this drop in adsorption was that it happened when additional molecules from the same locations made interaction with the adsorbent [43,44]. As a result, when the contact time is raised to 15 min, the interactions between the functional groups of the adsorbent and the antibiotic become stronger and achieve their maximum equilibrium adsorption capacity of 578.2 mg/g. Afterward, active sites were occupied, and the equilibrium state was approached. 

#### 3.2.4. The Initial Concentration

The influence of the initial concentration of the analyte on the adsorption capacity of MMIP was studied to better comprehend how different concentrations influence adsorbent capacity. The ciprofloxacin concentration was changed during the experiment, while the other parameters, such as the amount of adsorbent (0.003 g), contact duration (15 min), and pH of the solution (7.0), remained unchanged (Figure 7D). The adsorption capacity rose linearly as a function of the initial concentration to 400 mg/L. The antibiotic’s intensity climbed to 1035.4 when the starting concentration was raised from 100 to 400 mg/L. 

### 3.3. Isotherm Study

Ciprofloxacin’s interaction with MMIP was investigated using an adsorption isotherm analysis. The ability of a sorbent to hold sorbate is set theoretically. As a result, the value at which no further sorption may occur is referred to as the saturation value. Recently conducted studies often use Langmuir and Freundlich’s models to describe equilibrium adsorption isotherms and establish the maximum adsorption capacity. The Langmuir isotherm is the single-layer chemical absorption onto the adsorbent surface. Homogeneous adsorption is anticipated if all locations on the adsorbent surface interact with contaminants equally and with similar affinities [45]. The adsorption capacity is at its peak once contaminants form a complete monolayer on the adsorbent surface. In contrast, the multilayer adsorption of contaminants on the adsorbent’s heterogeneous surface is the basis of the Freundlich model. The following equations mathematically represent both the Langmuir (4) and Freundlich (5) models [46].
(4)$$\frac{C_{e}}{Q_{e}}=\frac{1}{K_{L}Q_{\max}}+\frac{1}{Q_{\max}}C_{e}$$
(5)$${\mathrm{LnQ}}_{e}={\mathrm{LnK}}_{f}+\frac{1}{n}{\mathrm{LnC}}_{e}$$
where C_e_ is the equilibrium concentration of ciprofloxacin (mg/L); Q_e_ and Q_max_ are the equilibrium and maximum adsorption capacity (mg/g), respectively; K_L_ (L/mg) and K_F_ (L/mg) are the Langmuir and Freundlich constants calculated from the plot between C_e_/Q_e_ and C_e_, and between Ln Q_e_ and Ln C_e_, respectively. n is a factor to determine the favorability of the adsorption process; when n > 1, the uptake of ciprofloxacin onto adsorbent is desirable at high concentrations. 

Figure 8A,B present the plots for the Langmuir and Freundlich isotherm models. Table 3 contains all of the relevant data from these two isotherms. According to the matching correlation coefficient (R^2^) of isotherm models, the Freundlich isotherm was shown to be more compatible with experimental data than the Langmuir isotherm. This illustrates multilayer antibiotic adsorption over the heterogeneous surface of MMIP. The created sorbent has a maximum ciprofloxacin adsorption capability of 1111.1 mg/g compared to other sorbents described recently (Table 3). This could be clarified by the polymer three-dimensional structure that can provide numerous reactive adsorption sites, including hydroxyl, amine, and carboxyl groups, as well as the presence of Fe_3_O_4_/CuO nanoparticles having high surface area. All of these factors can assist the adsorbent in successfully interacting with ciprofloxacin via electrostatic interactions, cation exchange, π–π interactions, and hydrogen bonding, and efficiently removing this antibiotic from an aqueous solution.

### 3.4. Adsorption Kinetics

The rate-limiting stage of the adsorption process, equilibrium time, and prospective adsorption processes can all be explored and evaluated using the useful tool of adsorption kinetics [54]. The ciprofloxacin adsorption rates on the MMIP were estimated in this work using the pseudo-first-order and pseudo-second-order models [55]. Equations (6) and (7), respectively, are used to mathematically quantify them [56,57].
(6)$$\mathrm{Ln}\left(Q_{e}-Q_{t}\right)={\mathrm{LnQ}}_{e}-\frac{k_{1}}{2.303}t$$
(7)$$\frac{t}{Q_{t}}=\frac{1}{k_{2}Q_{e}^{2}}+\frac{1}{Q_{e}}t$$
where Q_t_ (mg/g), and Q_e_ (mg/g) are the adsorption capacity (or amount of ciprofloxacin adsorbed onto sorbent) at time t and equilibrium, respectively. k_1_ (1/min) and k_2_ (g/mg⋅min) are the rate constants of the pseudo-first-order and pseudo-second-order, respectively. 

The kinetic linear plots and their calculated details are provided in both Table 4 and Figure 9A,B. By taking into account the correlation coefficients and the difference between the estimated Q_e_ and observed Q_e_ values of the two considered kinetic models, it can be seen that the pseudo-second-order model is better suited to explaining the ciprofloxacin adsorption kinetics on the MMIP.

### 3.5. Recovery and Reusability

In order to save money, time, and energy while protecting the environment, adsorbent recoverability and reusability are essential. Adsorbents having these characteristics therefore greatly minimize the difficulties involved in the disposal of old adsorbents and the production of new adsorbents [18,58]. The recoverability and regeneration of the MMIP sorbent were examined using three sequential cycles of adsorption and desorption tests. For this purpose, the ciprofloxacin adsorption test onto the MMIP was performed (reaction condition: pH = 7, MMIP dosage = 3 mg, contact time = 15 min, antibiotic initial concentration: 200 ppm, T = 298 K). Afterward, to desorb ciprofloxacin, the ciprofloxacin-loaded used MMIP was immersed in ethanol and agitated for 4 h at room temperature. Ciprofloxacin was therefore desorbed from fabricated MMIP and released into the solution, the adsorbent was separated using a magnet, washed many times with distilled water, and then dried to prepare for future adsorption/desorption investigations. Subsequently, using a UV-visible spectrophotometer, the amount of released ciprofloxacin in the elution medium was measured. As seen in Figure 10, after the third cycle, the adsorption percentage fell from 95.18 to 91.61%, and the desorption percentage from 94.01% to 89.86%. These results show that ciprofloxacin could be removed by the MMIP three times in a row without significantly weakening its ability to adsorb the antibiotic. 

### 3.6. Ciprofloxacin Adsorption Mechanism

The outstanding physicochemical properties of the MMIP adsorbent can be described as the high adsorption capacity of ciprofloxacin. The textural property, high surface area, and three-dimensional porous structure of the MMIP provided the convenient diffusion process and quick mass transfer of antibiotic molecules; the presence of a large number of reactive functional groups (-NH-, COOH, OH) of MMIP adsorbent, nanostructured Fe_3_O_4_/CuO, as well as possessing specific recognition ability, their high predetermined selectivity for ciprofloxacin antibiotic play significant role in the adsorption of this antibiotic from aquatic media. 

It is vital to know the solution chemistry information (pH, pKa, pHzc) to determine what chemical reactions could take place at certain pH levels. The pH of the solution affects the adsorbent–adsorbate interactions. According to the literature, ciprofloxacin shows two pKa values of 5.90 and 8.89 corresponding to the amine and carboxylic acid groups [59,60]. Ciprofloxacin has a cationic form because of the protonation of the amine group below pH 5. In addition, it shows zwitterionic properties between pH 5.9 and 8.9, and has an anionic form when the pH is greater than 8 due to the deprotonation of carboxylic acid. 

Furthermore, as mentioned above, the adsorption mechanism involves complicated interactions such as pore diffusion, hydrophobic interactions, and H-bonds (between the carboxyl groups of the ciprofloxacin molecules and the oxygen and nitrogen-containing groups of MMIP), π–π interactions, and electrostatic interactions.

Considering that the adsorbent exhibits a negative surface charge anywhere in the pH range of 1 to 5.59 of the solution, electrostatic interactions and cation exchange (with the protonated amine groups of ciprofloxacin and MMIP) often become active (Figure 11). The ciprofloxacin structure’s ability to form π–π interactions with adsorbents is due to the presence of the electron-rich benzene ring. When the pH of the solution is close to neutral (the pH range of 5.90 to 8.89) or the ciprofloxacin is present as a zwitterion, they come into play more forcefully. Similar pH ranges to those of the π–π interactions enable hydrogen bonding [61]. However, since ciprofloxacin was in anionic form after pH 8.9, it was repulsed by the negative surface of MMIP. In this case, a decrease in adsorption capacity was evidently observed after this pH value.

## 4. Conclusions

In the current study, MMIP was developed in three steps and applied as an adsorbent for ciprofloxacin removal. The prepared materials were characterized by FTIR, XRD, VSM, TGA, EDX, and FE-SEM analyses. Antibacterial experiments revealed that MMIP has an antibacterial effect against Escherichia coli (16 mm) and Staphylococcus aureus (22 mm). The pH value of 7, the adsorbent dosage of 3 mg, the contact time of 15 min, and the initial concentration of ciprofloxacin of 400 mg/L were the most effective parameters for the adsorption of ciprofloxacin onto the adsorbent. The experimental adsorption kinetics were successfully matched to pseudo-second-order and adsorption isotherm data were well fitted with the Freundlich model. The produced sorbent’s maximum adsorption capacity (Q_max_) for ciprofloxacin was determined to be 1111.1 mg/g. A significant portion of the ciprofloxacin was absorbed into the MMIP through electrostatic interactions, cation exchange, π–π interactions, electrostatic interactions, and hydrogen bonding. Adsorption–desorption trials demonstrated that the MMIP could be recovered and reused three times with no discernible loss in adsorption performance.

In terms of economic benefit, MMIP adsorbent is more expensive than adsorbents such as agricultural residues (rice husk, sawdust, walnut shells, etc.), and activated carbon, because it was fabricated from the graft copolymerization reaction of aniline and meta-phenylenediamine monomers with carboxymethyl dextrin in the presence of Fe_3_O_4_/CuO nanoparticles. Its superiority over the above adsorbents is its selectivity and high absorption capacity. On the other side, considering that the MMIP adsorbent is fabricated of carboxymethyl dextrin, biocompatible poly(aniline-*co*-m-phenylenediamine), and Fe_3_O_4_/CuO, therefore the MMIP adsorbent is a biocompatible compound with high adsorbability for the selective removal of ciprofloxacin from contaminated water environments.

## Figures and Tables

**Figure 1 nanomaterials-13-00489-f001:**
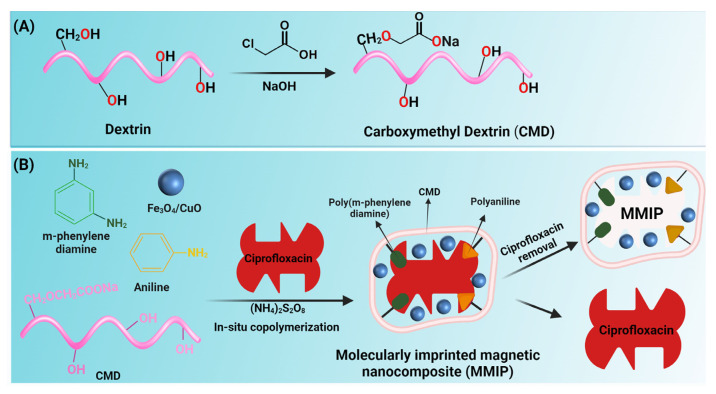
Schematic illustration of the synthesis of carboxymethyl dextrin (CMD) (**A**) and fabrication of magnetic molecularly imprinted polymer (MMIP) (**B**).

**Figure 2 nanomaterials-13-00489-f002:**
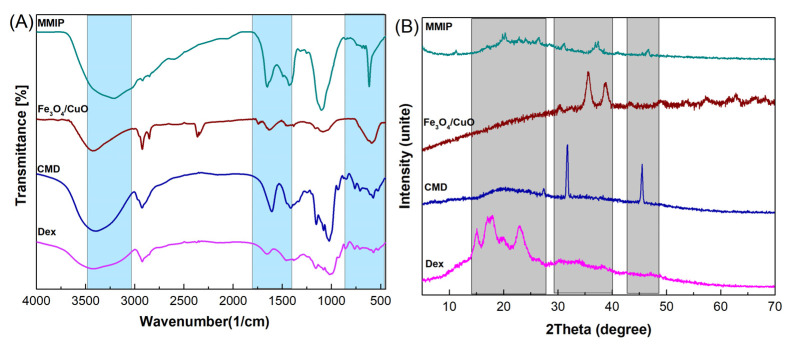
FTIR spectra (**A**) and XRD patterns (**B**) of dextrin (Dex), carboxymethyl dextrin (CMD), Fe_3_O_4_/CuO, and magnetic molecularly imprinted polymer (MMIP).

**Figure 3 nanomaterials-13-00489-f003:**
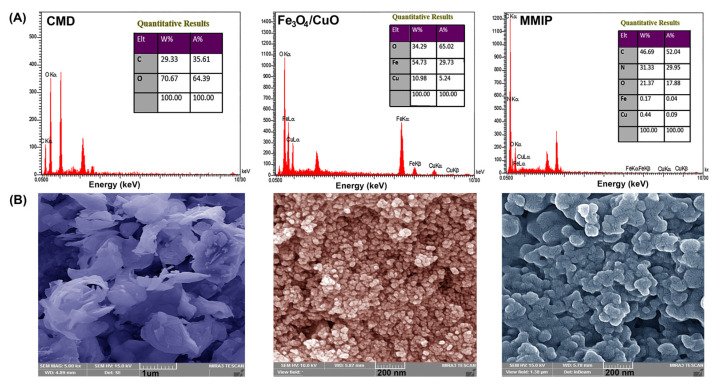
EDX spectra and tabulated data (**A**) and FESEM images (**B**) of carboxymethyl dextrin (CMD), Fe_3_O_4_/CuO, and magnetic molecularly imprinted polymer (MMIP).

**Figure 4 nanomaterials-13-00489-f004:**
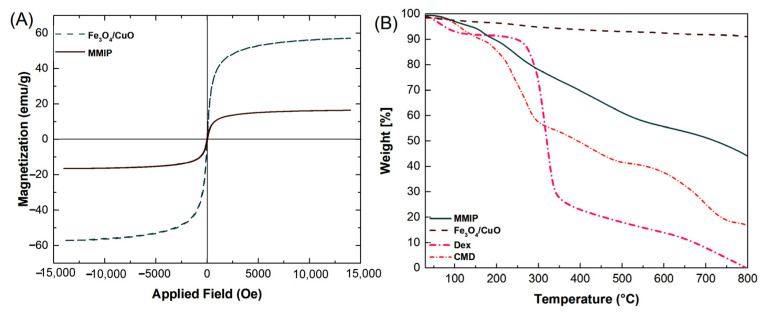
VSM curves of Fe_3_O_4_/CuO and magnetic molecularly imprinted polymer (MMIP) (**A**), and TG thermograms of dextrin (Dex), CMD, Fe_3_O_4_/CuO, and MMIP (**B**).

**Figure 5 nanomaterials-13-00489-f005:**
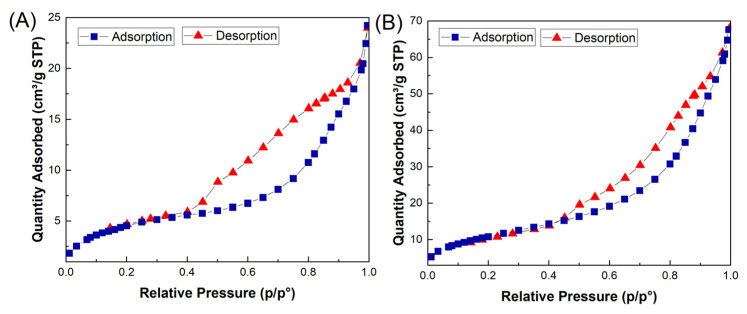
N2 adsorption/desorption isotherms of Fe3O4@CuO (**A**) and MMIP (**B**).

**Figure 6 nanomaterials-13-00489-f006:**
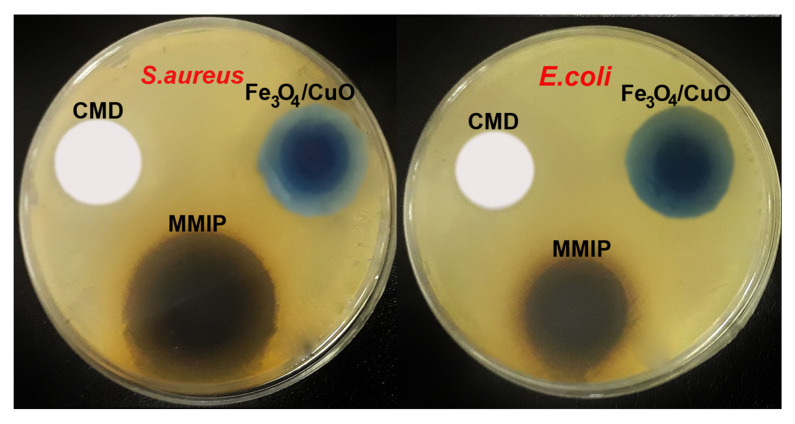
Cell culture of antibacterial activity of Fe_3_O_4_/CuO, carboxymethyl dextrin (CMD), and magnetic molecularly imprinted polymer (MMIP) against *Escherichia coli* (*E. coli*) and *Staphylococcus aureus* (*S. aureus*).

**Figure 7 nanomaterials-13-00489-f007:**
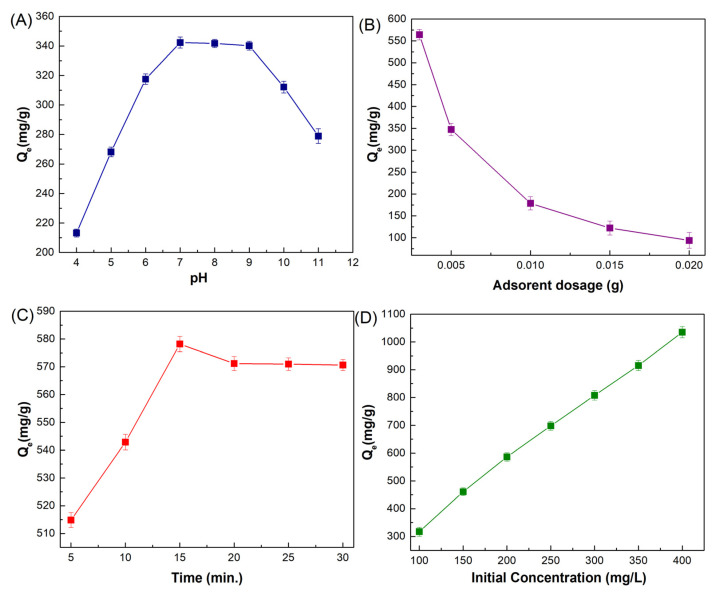
(**A**) Effect of solution pH (4–11) on adsorption capacity (mg/g) [MMIP dosage (5 mg), ciprofloxacin initial concentration (200 mg/L), V (0.01 L), Time (30 min), T (298 K)], (**B**) MMIP dosage (3–20 mg) on adsorption capacity (mg/g) [pH 7, ciprofloxacin initial concentration (200 mg/L), V (0.01 L), Time (30 min), T (298 K)], (**C**) contact time on adsorption capacity (mg/g) (5–30 min) [pH 7, MMIP dosage (0.003 g), ciprofloxacin initial concentration (200 mg/L), V (0.01 L), T (298 K)] and (**D**) ciprofloxacin initial concentration (100–400 mg/L) on adsorption capacity (mg/g) [pH 7, MMIP dosage (3 mg), V (0.01 L), Time (15 min), T (298 K)].

**Figure 8 nanomaterials-13-00489-f008:**
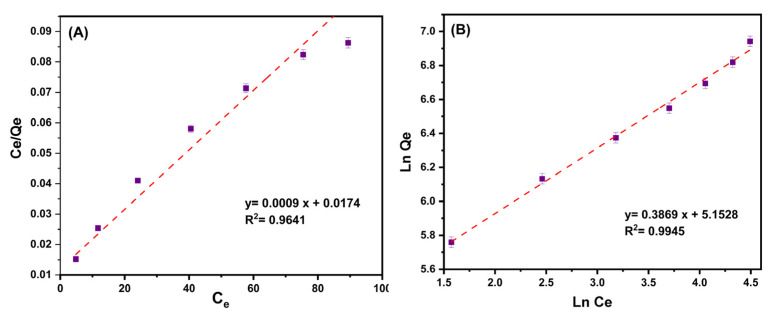
(**A**) The plot of the ratio of the equilibrium concentration of ciprofloxacin (C_e_, mg/L)/equilibrium adsorption capacity (Q_e_, mg/g) to the equilibrium concentration of ciprofloxacin (C_e_, mg/g) [Langmuir isotherm]. (**B**) The plot of the Ln of equilibrium adsorption capacity (Q_e_, mg/g) to the Ln of the equilibrium concentration of ciprofloxacin (C_e_, mg/g) [Freundlich isotherm]. Conditions: contact time (5–30 min), pH:7, MMIP dosage (3 mg), ciprofloxacin initial concentration (200 mg/L) and T (298 K).

**Figure 9 nanomaterials-13-00489-f009:**
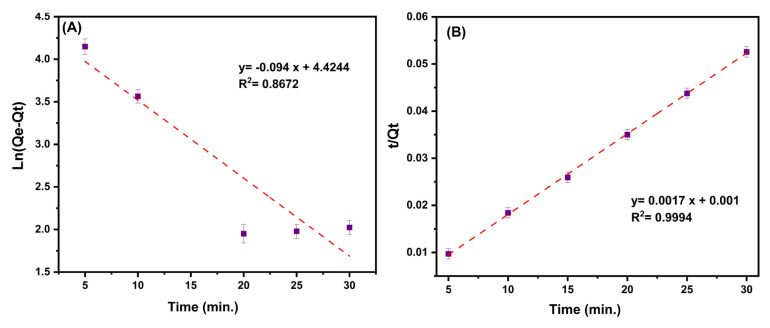
(**A**) The plot of the Ln of (equilibrium adsorption capacity (Q_e_, mg/g) minus adsorption capacity at time t (Q_t_, mg/g)) to time (min.) [Pseudo-first-order kinetic]. (**B**) The plot of the ratio of time (min)/adsorption capacity at time t (Q_t_, mg/g) to time (min.) [Pseudo-second-order kinetic]. Conditions: contact time (5–30 min), pH:7, MMIP dosage (3 mg), ciprofloxacin initial concentration (200 mg/L) and T (298 K).

**Figure 10 nanomaterials-13-00489-f010:**
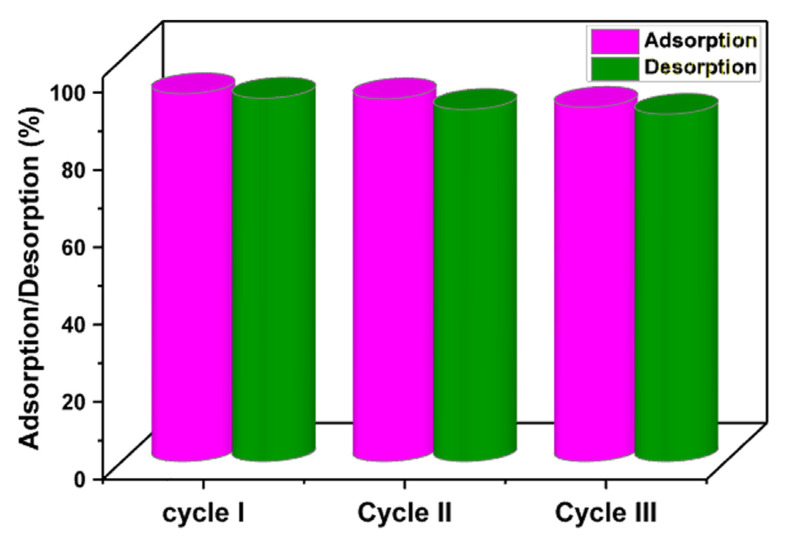
Reusability of the MMIP adsorption/desorption percentages of ciprofloxacin during three cycles. (Reaction conditions: contact time (15 min), solution pH:7, adsorbent dosage (3 mg), initial concentration (100 mg/L) solution volume: 10 mL. T (298 K)).

**Figure 11 nanomaterials-13-00489-f011:**
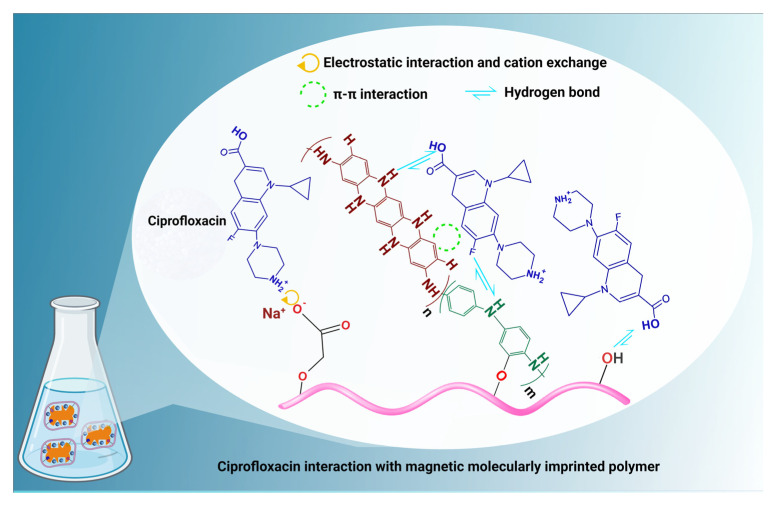
The suggested mechanism of ciprofloxacin adsorption on the MMIP.

**Table 1 nanomaterials-13-00489-t001:** Zeta potential values of MMIP at room temperature.

pH	Zeta Potential (mV)
6	−10.45
7	−12.72
8	−14.68

**Table 2 nanomaterials-13-00489-t002:** Antibacterial activity data of Fe_3_O_4_/CuO, CMD, and MMIP against *Escherichia coli* (*E. coli*) and *Staphylococcus aureus* (*S. aureus*). The diameter of the inhibitory zone of bacteria was measured in millimeters (mm).

Sample	*E. coli* (mm)	*S. aureus* (mm)
CMD	NE	NE
Fe_3_O_4_/CuO	18 ± 0.9	21 ± 0.5
MMIP	16 ± 1.1	22 ± 0.8
Gentamicin (10 μg/disk)	20.3 ± 1.1	24.3 ± 1.5
Chloramphenicol (30 μg/disk)	21.7 ± 1.5	23.7 ± 0.6

NE: No effect.

**Table 3 nanomaterials-13-00489-t003:** Adsorption capabilities of several adsorbents for ciprofloxacin.

Adsorbents	Q_max_ (mg/g)	pH	Tem. (°C)	Method of Determination	Ref.
Activated carbon from mangosteen peel	29.76	6	25	Langmuir	[7]
Montmorillonite clay	128.0	7	25	Temkin	[6]
Acid activated carbon from Prosopis juli-flora	250	4	25	Langmuir	[44]
Fe_3_O_4_ Nanoparticles	24.0	7	25	Freundlich	[45]
Konjac glucomannan/ZIF	812.4	7	30	Langmuir	[47]
Magnetic carboxymethyl chitosan	527.9	5	25	Langmuir	[48]
Fe_3_O_4_-SiO_2_-Schiff base	415.3	5.0	-	Freundlich	[49]
Al(III)-chelated cryogels of chitosan	390.0	7.5	23	Experiments	[50]
Modified alginate/graphene hydrogel	344.8	8.0	25	Langmuir	[51]
Sodium alginate/ĸ-carrageenan	291.6	5.1	25	Dubinin-Radushkevic	[52]
Magnetic chitosan/graphene oxide	282.9	5	25	Langmuir	[53]
MMIP	1111.1	7	25	Langmuir	Present work

**Table 4 nanomaterials-13-00489-t004:** Isotherm and kinetic constants, correlation coefficients, and statistical parameters, for adsorption of ciprofloxacin on the MMIP.

Model		Parameters	
Isotherm	Freundlich	K_F_ (L/mg)	172.915
		n	2.584647
		R^2^	0.9945
	Langmuir	Q_m_ (mg/g)	1111.11
		K_L_ (L/mg)	0.051724
		R^2^	0.9641
Kinetics	Pseudo-first-order	k_1_	0.094
		Q_e_ (mg/g)	83.46271
		Q_e experimental_ (mg/g)	578.2
		R^2^	0.8672
	Pseudo-second-order	k_2_	0.00289
		Q_e_ (mg/g)	588.2353
		Q_e experimental_ (mg/g)	578.2
		R^2^	0.9994

## Data Availability

Data available on request.

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
