# Peer review of "Molecularly Imprinted Magnetic Nanocomposite Based on Carboxymethyl Dextrin for Removal of Ciprofloxacin Antibiotic from Contaminated Water"

_nanomaterials, 2023, doi:10.3390/nano13030489_

Round 1

Reviewer 1 Report

Economic benefit and environmental friendliness of the prepared materials should be discussed in the manuscript.

Author Response

Economic benefit and environmental friendliness of the prepared materials should be discussed in the manuscript.

Response: Thank you very much for your valuable comment, in terms of economic benefit, MMIP adsorbent is more expensive than adsorbents such as agricultural residues (rice husk, sawdust, walnut shells, etc.), and activated carbon because it was fabricated from the graft copolymerization reaction of aniline and meta-phenylenediamine monomers with carboxymethyl dextrin in the presence of Fe3O4/CuO nanoparticles. Its superiority over the above adsorbents is its selectivity and high absorption capacity. On the other side, considering that the MMIP adsorbent is fabricated of carboxymethyl dextrin, biocompatible poly (aniline-co-m-phenylenediamine), and Fe3O4/CuO, therefore the MMIP adsorbent is a biocompatible compound with high adsorbability for the selective removal of ciprofloxacin from contaminated water environments.

The above paragraph has been added to the “Conclusion” section.

Reviewer 2 Report

The manuscript reports on the synthesis of the hybrid molecularly imprinted nanocomposite formed of carboxymethyl dextrin, polymeric oxidized aminobenzenes, and Fe3O4/CuO nanoparticles. The prepared material was characterized by various physical methods. The activity of the material as a sorbent for ciprofloxacin antibiotic in water solutions was studied and mathematically analyzed.

The paper is informative and well-written. It can be published after solving several issues.

1) Preparation of Fe3O4/CuO hybrid nanoparticles (line 142). The authors report that hybrid Fe3O4/CuO nanoparticles were formed from the Fe3O4 and CuO powders in a weakly alkaline hot water medium by means of ultrasonic treatment. What is the chemical nature of the formation of mixed Fe-Cu oxide particles from the separate metal oxides? Which processes take place?

2) In the heading of Table 1, there should be an explanation of what those "mms" mean since it is not clear from the nearby text.

3) Line 411 - "Table 2" should be instead of "Table 1".

4) In Figure 4 caption, there should be noted, what is qe on y-axes.

5) In Figure 6B, a fragment of 2,5-diaminophenol in some stacking interaction with ciprofloxacin molecule is depicted. Could the fragment of 2,5-diaminophenol be formed in the hybrid material when aniline and m-phenylenediamine were the starting substances? It seems doubtful.

Reviewer 3 Report

The results are interesting and some reasonable explanation is provided. It is acceptable with revision. However, several modifications are required as follows:

1.      The adsorptive performance of the sample should be compared with commercial ones to highlight its significance.

2.      The BET surface areas of the samples should be measured and analyzed.

3.      TEM characterizations of the sample are necessary and valuable to this work.

4.      The advanced techniques for the removal of antibiotics should be introduced to keep the latest research trends. e.g.: Chin. J. Catal., 2022, 43, 2652–2664, Advanced Fiber Materials 2022, 4, 1620–1631, Sep. Purif. Technol., 2023, 304, 122401.

5.      The mechanism for the high adsorption ability of the sample should be investigated and discussed in detail.

6.      How about the adsorption capability of the sample in real waters (e.g. river water).

7.      The surface charge of the sample under different pH should be experimentally revealed to illustrate the adsorption ability variations under these conditions.

8.      The adsorption performance of the sample under pH= 9 or 11 should be investigated.

9.      The regeneration procedure of the sample should be provided in detail.

Reviewer 4 Report

The title is strange. What the results of this study say is that the manufactured MMIP can be used for antibacterial activity and can also be used for detecting antibiotics in aqueous systems, but the title is difficult to convey this.

An explanation of the MMIP fabrication process is shown in Figure 1A,B, but it is poor. An improved picture is needed.

Figures 1C and D should not be merged with Figures 1A and B, but should be separated into other figures.

Figure 3A,B should be separated from Figure 3C.

How do the MMIPs have the antibacterial effect?

Figure 4 caption should be more detailed. Missing information about the concentration of Ciprofloxacin tested. Missing information about what the y-axis qe is. Missing information about what each of the x-axis parameters (pH, dosage, time, and initial concentration) means. In addition, it is necessary to summarize the main points of what the author wants to say in conclusion in Figure 4 by tying the four graphs together.

In Table 2, in addition to Qmax information, limit of detection (LOD) comparison is also required.

In Figure 5, information of the x-axis and y-axis parameters of each graph are ambiguous. For Figure 5c, a linear fit is not appropriate. Table 3 should be modified accordingly.

What Figure 6 is trying to say is reusability. real sample test with drinking water should be needed, not DI water. How can the MMIP sprayed in drinking water be retrieve when its lifespan is over? For the use of MMIP, are there any problems with secondary water pollution, cytotoxicity, and bioconcentration?

Authors used the expression "synergistic mechanism" in B, but this expression is not appropriate. And the figure of Figure 6b should be separated from Figure 6A and rather moved to Figure 1.

Important references were missing. For example,

https://link.springer.com/article/10.1007/s00604-019-3985-5

Introduction should be revised by adding those previous works.

Round 2

Reviewer 3 Report

The authors have carefully revised the manuscript and it can be accepted now.

Reviewer 4 Report

The revised manuscript can be accepted.